# Selection of Meat Inspection Data for an Animal Welfare Index in Cattle and Pigs in Denmark

**DOI:** 10.3390/ani7120094

**Published:** 2017-12-06

**Authors:** Søren Saxmose Nielsen, Matthew James Denwood, Björn Forkman, Hans Houe

**Affiliations:** Section for Animal Welfare and Disease Control, Department of Veterinary and Animal Sciences, University of Copenhagen, DK-1870 Frederiksberg C, Denmark; md@sund.ku.dk (M.J.D.); bjf@sund.ku.dk (B.F.); houe@sund.ku.dk (H.H.)

**Keywords:** abattoir recordings, animal welfare, cattle, meat inspection, pigs, slaughterhouse

## Abstract

**Simple Summary:**

Despite being important to the general public, the monitoring of animal welfare is not systematic. The Danish political parties agreed in 2012 to establish national animal welfare indices for cattle and pigs, and here we assess the potential for using data from the systematic meat inspection to contribute to such indices. We demonstrate that although a number of recordings may be relevant for animal welfare, differences in recording practices between slaughterhouses can be so large that correction is not deemed feasible. For example, significant differences in tail fractures in pigs and sows were recorded between abattoirs, despite the fact that this condition should be easier to diagnose compared to e.g., the more consistently recorded “chronic arthritis” in cows. The study findings suggest that some recordings may be useful for inclusion in animal welfare indices, but that their relevance should be assessed along with the recording practices if included. Furthermore, factors such as appropriate behaviour are also important to monitor as part of the welfare of both cattle and pigs.

**Abstract:**

National welfare indices of cattle and pigs are constructed in Denmark, and meat inspection data may be used to contribute to these. We select potentially welfare-relevant abattoir recordings and assess the sources of variation within these with a view towards inclusion in the indices. Meat inspection codes were pre-selected based on expert judgement of having potential animal welfare relevance. Random effects logistic regression was then used to determine the magnitude of variation derived at the level of the farm or abattoir, of which farm variation might be associated with welfare, whereas abattoir variation is most likely caused by differences in recording practices. Codes were excluded for use in the indices based on poor model fit or a large abattoir effect. There was a large abattoir effect for most of the codes modelled and these codes were deemed to be not appropriate to be carried forward to the welfare index. A few were found to be potentially useful for a welfare index: Eight for slaughter pigs, 15 for sows, five for cattle <18 months of age, and six for older cattle. The absolute accuracy of each code/combination could not be assessed, only the relative variation between farms and abattoirs.

## 1. Introduction

In 2012, a joint agreement between the political parties represented in the Danish parliament decided to establish animal welfare indices [1]. The purpose of the development of national indices for cattle and pigs was to enable surveillance of the state of animal welfare nationally and in the longer term decide areas where animal welfare can be improved. Animal welfare is, however, a multifactorial concept with different stakeholders traditionally thought to emphasise different aspects [2,3,4]. To create an index that is transparent it was decided to choose a hedonistic approach to animal welfare. This approach places the emphasis on the experiences of the animal [5], with the consequence that e.g., disease or reduced growth are only taken into account if they have an impact on the affective state of the animal. This is the same approach as the one taken in the EU-project Welfare Quality [6]. The indices were to be constructed using farm visits, but in order to make the monitoring as efficient and cheap as possible, there was also a desire to include register data whenever possible. 

Meat inspection is carried out routinely on all cattle and pigs carcasses according to legislation from EU and Denmark [7,8] in order to safeguard food and animal welfare at slaughter. The meat inspection data may also be used for purposes such as creation of an index of animal welfare. A number of challenges exist prior to such use. For example, all meat inspection parameters recorded for food safety reasons are not necessarily relevant in relation to animal welfare at the farm, and some are related to acute disease conditions, which may have occurred during transport, and some are fairly non-specific recordings. Furthermore, differences in recording practices and thresholds may differ between slaughterhouses [9,10,11], which may result in differences in sensitivity and specificity of the meat inspection data in relation to the intended target conditions between the slaughterhouses. Finally, rare conditions may be difficult to appraise statistically, although they are of sufficient severity to highly motivate inclusion in a welfare index. 

The objectives of the present study were to provide a statistical assessment of meat inspection data to (a) select codes of relevance to an animal welfare index based on prevalence and welfare impact; (b) assess the contribution of each slaughterhouse on the variation in prevalence of each relevant meat inspection variable; and (c) provide estimates of a correction factor for each slaughterhouse for each of the relevant meat inspection code.

The assessments were done separately for cattle aged <18 months (hereafter denoted ‘calf’), cattle aged ≥18 months (hereafter denoted ‘cow’), slaughter pigs, and sows.

## 2. Materials and Methods

Meat inspection data for 2012 were provided by the Danish Veterinary and Food Administration (Glostrup, Denmark) and used for the data analyses. The meat inspections are done by official technicians as laid down in the EU legislation [7]. A specific protocol is given in a government circular [8], according to which an official veterinarian has the overall responsibility of the recording as specified in the EU legislation. Observations are recorded electronically at the carcass inspection station and verified by government veterinarians and uploaded to a meat inspection database located with the Danish Food and Agricultural Council (Axelborg, Copenhagen V, Denmark). The data were summarised into the number of animals slaughtered and prevalence of code, for each combination of farm of origin, abattoir, animal type (pig, sow, calf, cow), and slaughter date. Data were provided from all major pig (*n* = 9) and sow (*n* = 3) abattoirs, including 5381 pig farms and 1781 sow farms. Slaughterhouses processing relatively few cattle were excluded, i.e., all slaughterhouses with less than 10,000 cattle slaughtered in 2012 were not included in the following analyses. This resulted in data from eight slaughterhouses being used, with a total of 10,718 farms providing data for cows and 7019 farms providing data on calves. Cows and calves were slaughtered in the same abattoirs, whereas pigs and sows were slaughtered in separate plants. Due to the purpose of the study, namely to create an index reported annually, observations from all dates were then combined at the level of farm, abattoir, code and animal type. This was referred to as a “batch”, i.e., a batch consisted of the number of pigs, sows, cattle <18 months, or cattle ≥18 months of age slaughtered at a specific abattoir from a specific farm within 2012. 

### 2.1. Exclusion of Codes

Some irrelevant “commercial codes” (such as information about contamination, missing organs and slaughter line issues) were excluded from the data. Specific meat inspection codes were also excluded where they were not deemed relevant to the purpose of the study, which was to assess changes in on-farm welfare of cattle and pigs, excluding transport to the abattoir and slaughter. Consequently, codes were excluded due to (a) possibly being related to transport; (b) acute conditions, which could have occurred during transport; (c) central nervous system (CNS) conditions, while they are relatively unspecific and difficult to assess at the abattoir; (d) not related to animal welfare (when using the hedonistic definition mentioned previously); and (e) being non-specific conditions. Further, codes were excluded if they had a low prevalence combined with a low impact on welfare.

All individual codes were 3-digit (listed in Appendix A). Codes that were judged to be equivocations as far as animal welfare was concerned were collapsed into a single category. For example, all codes associated to included liver conditions in cattle were collapsed (374, 375, 377, 379, 381 to 374375377379381), and abscesses were collapsed to 570577580584585 irrespective if they occurred in the front part (570), mid-part (577), rear part (580), extremities (584) or head (585). If an animal had one of these conditions, it was classified as having the condition. The decisions were based on consensus between three of the authors (Hans Houe, Søren Saxmose Nielsen, Björn Forkman) and other experts (Sine Andreassen and Anne Marie Michelsen). See Appendix A, Table A1 (pigs) and Table A2 (cattle) for specific descriptions of the individual codes. 

### 2.2. Estimation of Abattoir Effects for Each Code and Category

Random effects logistic regression using R [12] was done as described in detail in Denwood et al. [13]. Briefly, the random effect logistic regression models were fitted using the glmer-function in the lme4 package in R [14]. The random effects model with binomial response was used to assess the relative variance explained by the farm of origin, abattoir, and residual extra-binomial variance at the level of “batch” observation (interaction of Farm and Abattoir). Models were fitted separately for each combination of animal type and code. To assess if abattoir and farm effects were present, the statistical significance of the random effects of Abattoir and Farm were individually tested using a numerical approach as described by Lewis et al. [15] and Denwood et al. [13]—where these were not deemed to be significant, they were removed. 

Animal type/code combinations with either fewer than 50 positive batches, or no batches with more than 1 positive animal, were not analysed using the random effects model (where batch as previously defined is the number of pigs, sows or cattle of a given type slaughtered at a specific abattoir from a specific farm). These datasets contain insufficient information for the random effects results to be numerically stable. Model fit was assessed against the distribution of deviance statistics from data generated using the fitted model. The general form of the model is as follows:Logit (p*_i_*) = A + B*_i_* + C*_f_* + D*_k_*
Y*_i_* ~ Binomial(p*_i_*, N*_i_*)
where the subscript *i* denotes each observed combination of farm and abattoir, *f* denotes the farm associated with batch *i*, and *k* denotes the abattoir associated with batch *i*. The explanatory variables consist of a common intercept A and random effect of batch B (which were included for every model), and random effects of farm C and abattoir D (which were tested for significance as discussed above). The response variable Y*_i_* (the number of observed positive recordings for batch *i*) was described using a Binomial distribution, according to the fitted probability p*_i_* and total number of recordings N*_i_*. The 95% confidence intervals for the estimates within the random effects associated with each farm and abattoir were generated using a parametric bootstrap approach. We note that a subset of this data has already been presented to illustrate the statistical methodology developed to analyse the data [13], but here we consider the welfare implications of the analyses rather than the statistical methods themselves, and also widen the scope to include both pigs and cattle.

The resulting random effect coefficients (on the logit scale) for codes where a statistically significant abattoir effect was identified were subsequently used to divide the modelled codes into those where: (i) correction of slaughterhouse effects might be useful for further use of the code; (ii) correction for slaughterhouse effect would be deemed controversial; and (iii) correction would be deemed inappropriate. For the former, random effect coefficients of between −1 and 1 were deemed potentially useful to generate correction factors, (under the assumption that they had acceptable sensitivity and specificity; this assumption is not assessed in this article). Any correction should be done on the logit scale, but for explanatory purposes, a random effect coefficient of 1 on the logit scale corresponds to a correction of approximately 2.7 times the average, and a random effect coefficient of −1 corresponds to a correction of 0.37 times the average (these approximations are only accurate for prevalences <20%; otherwise a correction has to be done on the logit scale). For larger random effects estimates it is likely that there is a systematic difference in recording procedure between slaughterhouses, so if the absolute random effect coefficient was between 1 and 2 (prevalences +/−2.7 to 7.4 times different between the abattoirs), then correction was deemed questionable; and if >2 then it was deemed inappropriate.

## 3. Results

### 3.1. Code Selection

The pig and sow data originally included 76 non-commercial meat inspection codes, while codes 101, 111, 113, 114, 115, 451, 501, 535, 542, 901, 903, 904 were excluded possibly being transport-related, codes 221, 287, 320, 350, 371, 402, 431, 471, 504, 506, 531, 551, 608 where considered possibly acute conditions, code 203 is a central nervous system diagnosis, and codes 181, 382, 385, 565, 815, 829, 890 were not deemed animal welfare related, while codes 602 and 603 are non-specific condition and 572 and 634 had a very low prevalence with likely low impact on animal welfare. A total of 20 individual codes and 8 categories thus remained (Table 1 and Table 2).

The cattle data originally included 84 non-commercial meat inspection codes while codes 101, 113, 115, 451, 535, 536, 537, 538, 542 were excluded as transport related, codes 133, 221, 258, 287, 320, 334, 350, 365, 371, 402, 431, 471, 501, 504, 506, 531 as acute conditions, 204 and 304 as central nervous system conditions, 119, 181, 382, 524, 551, 560, 561, 562, 563, 565, 815, 890 as not related to animal welfare, and 335 was considered non-specific. This resulted in the 19 codes and 9 categories listed in Table 3 and Table 4.

### 3.2. Descriptive Statistics

Prevalence for each code and code combination for slaughter pigs and sows are given in Table 1 and Table 2, respectively. Prevalence for each code and code combination for cattle are given in Table 3 and Table 4.

### 3.3. Random Effects Logistic Regression

#### 3.3.1. Pig and Sow Data

Eleven codes were removed from each of the pig and sow data because of poor model fit, which was primarily as a result of low numbers of observations (Table 5). Of the remaining 31 codes or combinations for each animal group, there was evidence of Abattoir-only variance for two sow-codes, Farm-only variance for five of each sow and slaughter pig codes, and both sources of variance for 33 combinations (eight combinations had neither random effect term fitted). For example, for code 120 in pigs, the variance effect due to abattoirs was 0.29, the farm effect was 0.38 and the residual 0.15. Thus, the farm effect was biggest, but there was still considerable difference between slaughterhouses (all abattoir and farm random effects terms presented are statistically significant). However for sows, the slaughterhouse effect appeared to be largest (0.36 vs. 0.26) meaning that the slaughterhouse effect seemed to be larger than that of disease. Figure 1 shows a graphical summary of the random effects. 

#### 3.3.2. Calf and Cow Data

Twenty-four and 19 codes were removed from the calf and cow datasets, respectively due to no and poor model fit, with 20 codes in calves and 25 codes cows producing acceptable model fits (Table 6). Of the remaining combinations, there was evidence of Abattoir-only variance for 8, Farm-only variance for five, and both levels of variance for 13 combinations (12 combinations had neither random effect term fitted). A summary graph illustrating the results is shown in Figure 2. 

There is substantially more agreement for the abattoir random effect estimates for the cattle data than for the pig data. However, there is still some variation in the magnitude of random effects estimates between codes, suggesting that caution should be taken when interpreting codes. There is a striking similarity between the estimates produced for calf and cow data, especially for disease codes 271289, 412, 570577580584585 and 602604.

#### 3.3.3. Pigs, Sows, and Cattle Combined

There was an abattoir effect for (a) all 31 modelled slaughter pig codes (12 individual and five code categories); (b) 26 of 31 modelled sow codes (12 individual and five categories); (c) all 21 modelled codes in cattle <18 months (four individual and five categories); and (d) 26 of 27 modelled adult cattle codes (seven individual and six categories) (Table 7). Including both the codes and categories with an abattoir effect and those without, (a) four codes and four categories (15 codes in total) were deemed potentially useful in pigs; (b) 10 codes and five categories (23 codes in total) were deemed potentially useful in sows; (c) two codes and three categories (14 codes in total) were deemed potentially useful in cattle <18 months; and (d) five categories (17 codes in total) were deemed potentially useful in cattle ≥18 months of age (Table 7). The potentially useful codes with descriptions are listed in Table 8.

## 4. Discussion

This study provides estimates of the differences in meat inspection recording due to farm and abattoir effect for a selection of meat inspection codes from three sow, nine pig and eight cattle abattoirs. “Farm”-associated variation is considered to be due to differences in health or welfare conditions at farms, whereas “abattoir”-associated variation might be considered to occur due to differences in recording at different abattoirs. However, it should be noted that a proportion of this variation may also be due to any systematic difference in the average prevalence of disease between the subsets of farms that primarily send animals to a specific abattoir for slaughter.

Among 76 meat inspection codes in pigs and sows, 42 were used as single codes or in categories in the random effect analyses. Thirty-one codes could be modelled in pig abattoirs and 31 could be modelled in sow abattoirs, but the codes were not exactly the same because different conditions were more prevalent in some types of animals than others. A farm and an abattoir type effect existed for all of these 31 pig codes and an abattoir effect existed for all but six codes/categories (132 (skinny), 230 (endocarditis), 379381 (liver conditions) and 600601 (tail-bite or association infection) in sows.

Among 84 meat inspection codes in cattle, 44 were used as single codes or in categories. Twenty codes could be modelled for calves and 25 for adult cattle. There was a significant abattoir effect for all but one code (532 (chronic arthritis or arthrosis)) in adult cattle.

There does not seem to be a great deal of consistency in abattoir effects between different disease codes in either pigs or sows, although some pairs of codes (for example Codes 336 (gastric ulcers) and 120 (circulatory affection) in pigs) do show some agreement. A similar analysis conducted using 2013 and 2014 data also revealed some variation from year to year (data not shown). There are also substantial differences in the estimate for the variance partition due to abattoir between disease codes, indicating that it is not likely to be feasible to use a single correction factor for all disease codes, if correction factors were to be used to even out the observed bias. For example, abattoir S10 was above average for five, and below for 11 codes and code categories, while abattoir S5 was above average for 13 and below average for seven codes and code categories (Figure 1). The individual random effect estimate for each abattoir can be interpreted as the effect of the abattoir on the reported prevalence of each code after accounting for differences between farms. This effect is relative to an 'average' abattoir with an effect size of 0 (i.e., a random effects estimate), so it can be used as the basis of a correction factor by multiplying the estimate by −1 and adding this to the logit of the average prevalence to come up with an expected logit prevalence at each abattoir. For prevalence <20%, which is true of almost all relevant slaughter codes, this can be reasonably approximated using the exponent of the abattoir effect multiplied by the observed prevalence. Obviously these estimates are conditional on the 2012 data being fully representative of future observations, and no effect of date/time of year has been accounted for so the correction factors can only safely be applied to a dataset representing a full calendar year of observations. 

For some codes, the results presented here suggest a considerable and significant difference in recording levels between abattoirs. The magnitude of the differences between abattoirs was most frequently observed in the range –1 to 1 (on the logit scale), but for some codes and categories the differences were somewhat larger or substantially larger (Table 7). For these codes, there would seem to be some structural differences in the recording procedures, and consequently applying a simple correction factor without addressing understanding of the major underlying differences in recording procedure may not be a sensible or viable approach. When the differences are smaller, then use of a correction factor to “even out” small variations between abattoirs may be useful to allow a more robust comparison of observed farm prevalence. There are some farms that only use one slaughterhouse, which should not be a problem for slaughterhouse effects, as slaughterhouses always have more than one farm. However, it constitutes a challenge that batch and farm effects confound each other for some farms, where a farm has a single batch and therefore two random effect levels for a single observation. Therefore, we may have challenges in separating the farm and batch effect, and interpretation of the data should focus on the abattoir effect, not the any potential farm-effect. It is also important to note that the random effects components presented are only estimates, and represent only indications of relative differences between welfare indicators and between abattoir and farm effects. Although it is theoretically possible to obtain confidence intervals for these via a procedure such as parametric bootstrapping, this is computationally impossible for this dataset. We also note the increased potential for shrinkage for the abattoir random effect relative to that for farm due to the large difference in the number of abattoirs (eight for cattle, nine for pigs and three for sows) vs. farms (10,718 farms for adult cattle, 7019 farms for calves, 5381 farms for pigs and 1781 farms for sows). This means that the variation between abattoirs is likely to be somewhat underestimated relative to that between farms. However, this does not affect our conclusions because of the focus on the abattoirs, not the farms.

Table 8 provides a list of meat inspection codes and descriptions for those codes and categories where there was no detected abattoir effect or where the effect was within −1 and 1 on the logit scale, i.e., they were within 2.7 times higher or lower than the mean prevalence. The listed conditions all have some relation to animal welfare, but we have refrained from specifying how much they would eventually contribute. This is dealt with in the weighting and aggregation in other parts of the main project. Furthermore, this study does not inform if the conditions are recorded accurately. Differences in accuracy of recording practices are likely to be the main cause of differences between slaughterhouses resulting in the high abattoir effects; differences in recording accuracy has also been demonstrated for clinical recordings [16]. It can be speculated that the conditions not recorded by some meat inspectors are those that are considered to be least severe. There are no data in the present study to suggest so, but it could be object of speculation. The conditions listed in Table 8 are those that are more specific and this supports the notion that they may be more accurately recorded. However, a condition such as gastric ulcers (code 336) in pigs might also be considered fairly specific and easy to diagnose, but there is still quite a large difference between the slaughterhouses. Chronic pericarditis (code 222) is also fairly specific and appears to be recorded relatively similarly in adult cattle across slaughterhouses, but this is not the case in pigs and sows, where the prevalence can still be high in some slaughterhouses (e.g., 5.1% in pigs in S1) but not in others (0.006% pigs in S6). Use of the data would depend on a farm-effect, because this effect should reflect the differences in the conditions. 

A number of additional requirements are necessary if the data should be used for national animal welfare monitoring. Firstly, the recordings should measure animal welfare with some level of accuracy, the recordings should be objective, consistent over time and feasible to implement. A basic assumption for use of the correction factors is that the time period used is representative. The recording level can differ within the same abattoir over time as we have previously demonstrated [10]. However, if the correction factors are updated regularly, e.g., annually, then this is only of minor importance. A more important assumption is that farmers do no send specific pigs (with e.g., higher or lower perceived prevalence of welfare-related conditions) to specific slaughterhouses, which would mean that true prevalence is made artificially high or low by the correction. Another example may be if certain types of pigs associated with particularly good or bad welfare are predominantly slaughtered at a particular slaughterhouse. For example, organic pigs are often slaughtered at specific slaughterhouses such as S4, and they may have different levels of disease. This could lead to e.g., a high prevalence at the abattoir slaughtering these specific pigs. Slaughterhouse S4 had a higher prevalence of codes 131 (emaciated), 132 (skinny), 222 (chronic pericarditis), 361 (hernias) and 505507 (healed tail and rib fractures), none of which is likely to be associated specifically to organic production. Farmers probably do not send pigs to slaughterhouses in any kind of balanced way, but we have no possible means to estimate this at the moment. For now, we have to accept that we cannot differentiate low slaughterhouse sensitivity from a slaughterhouse, where everyone sends the healthy animals, i.e., we assume that the distribution of true disease is random between slaughterhouses, which may be nonsense due to spatial effects of disease prevalence for some conditions, but not for others. However, it is not really possible to deem based on the data at hand. It should be noted that approximately 20% of sows are slaughtered in abattoirs not included in this study, while this is the case for less than 1% of slaughter pigs. Almost all cattle slaughtered in Denmark during 2012 were also included. However, it was not possible to correct for any imbalances in the data, which are observational in nature. The next steps in any data aggregation are also important but will not be covered here, as they are beyond the scope of the present paper. A thorough analysis has been included and published in a report from the Danish Veterinary and Food Administration including technical appendices [17]. 

Use of the data for an animal welfare index would also presume that all animals are slaughtered in Denmark. A high proportion of piglets are exported, and the number of sows slaughtered outside Denmark is also significant. Such animals would therefore not contribute to an animal welfare index.

## 5. Conclusions

We recommend to proceed with the codes and categories listed in Table 8, while they have some relation to animal welfare and differences in recording between abattoirs seem minimal to moderate. However, the accuracy of recording has not been assessed, and the magnitude of the relation to animal welfare has not been assessed either, although a qualitative assessment has been done. A full assessment would not be feasible. The codes and categories not included in Table 8 should not be used without further addressing differences between slaughterhouses. Last but not least, if the codes and categories are included in indices used for national governance, it should be recalled they are numeric simplifications of complex concepts [18].

## Figures and Tables

**Figure 1 animals-07-00094-f001:**
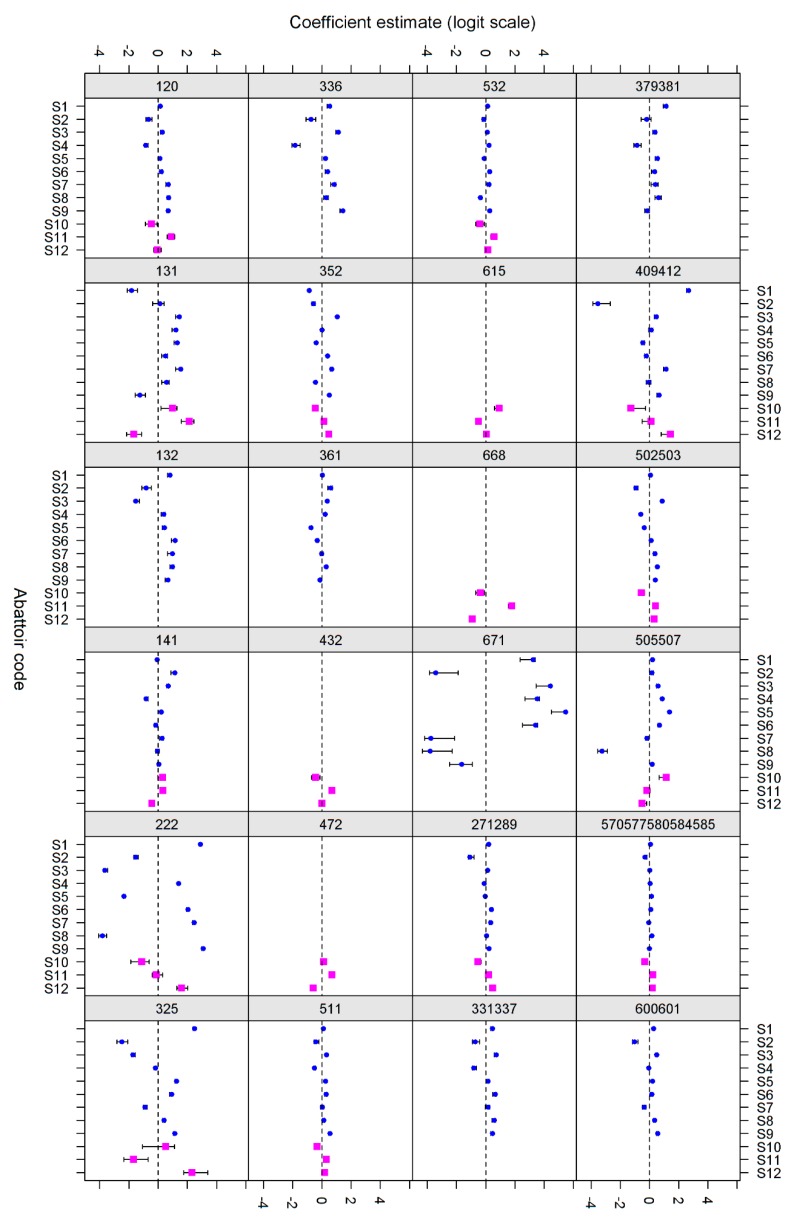
Individual estimates for the variance partition effect of each abattoir (95% confidence intervals shown as bars) for each code in pigs (S1–S9, blue) and sows (S10–S12, pink).

**Figure 2 animals-07-00094-f002:**
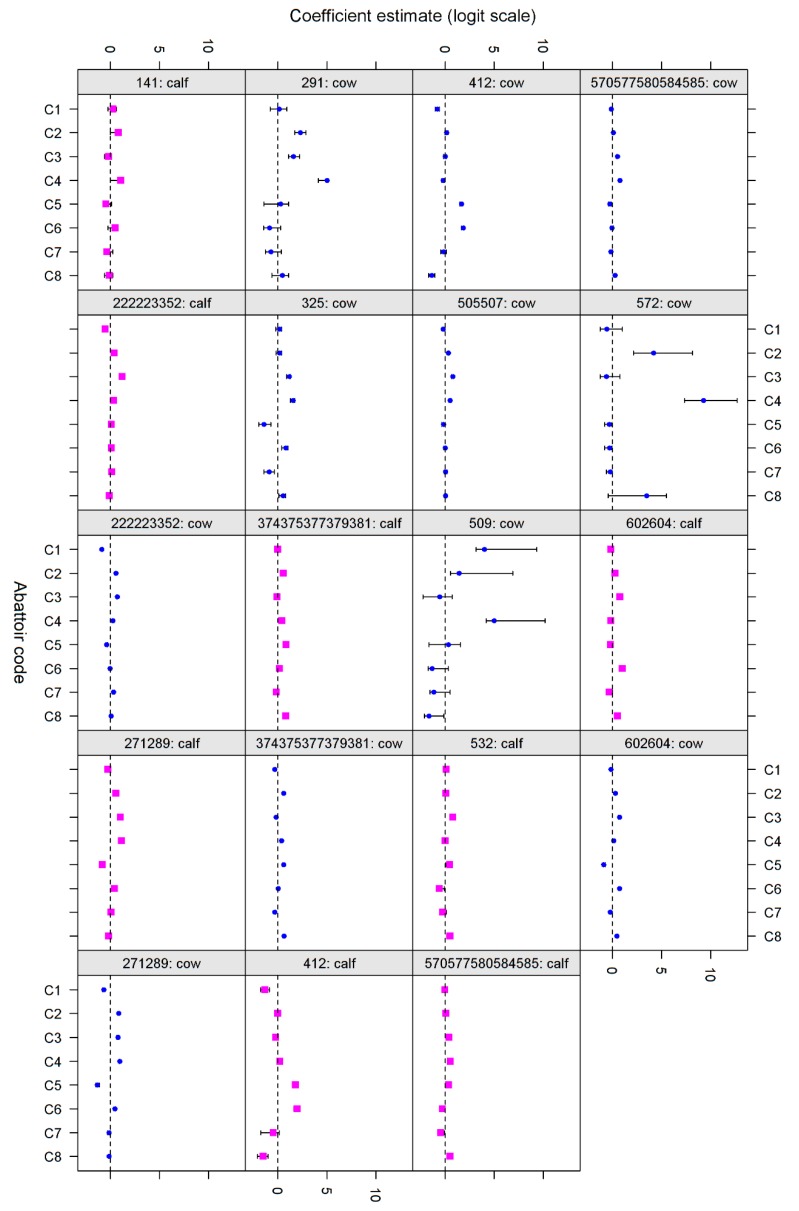
Individual estimates for the effect of each abattoir (95% confidence intervals shown as bars) for each code in calves (pink) and adult cattle (blue).

**Table 1 animals-07-00094-t001:** Prevalence (number and %) of selected slaughter recording codes in slaughter pigs slaughtered at the nine largest slaughterhouses (S1–S9) in Denmark in 2012.

Code/Category	S1		S2		S3		S4		S5		S6		S7		S8		S9	
	No.	%	No.	%	No.	%	No.	%	No.	%	No.	%	No.	%	No.	%	No.	%
120	1076	0.038	76	0.017	1177	0.043	207	0.014	450	0.066	483	0.074	1673	0.075	1744	0.038	736	0.042
131	15	0.001	24	0.005	414	0.015	181	0.012	127	0.019	42	0.006	22	0.001	640	0.014	105	0.006
132	999	0.035	26	0.006	77	0.003	307	0.021	267	0.039	264	0.040	701	0.031	997	0.022	820	0.047
141	1130	0.040	695	0.155	2442	0.089	274	0.019	370	0.054	256	0.039	974	0.044	2460	0.054	674	0.039
222	144,357	5.115	269	0.060	186	0.007	17,200	1.179	21,771	3.202	38	0.006	124,856	5.599	1199	0.026	35,780	2.061
230	108	0.004	9	0.002	59	0.002	71	0.005	23	0.003	14	0.002	123	0.006	50	0.001	76	0.004
250	32	0.001	0	0.000	19	0.001	307	0.021	99	0.015	10	0.002	315	0.014	635	0.014	65	0.004
258	606	0.021	163	0.036	558	0.020	81	0.006	179	0.026	7	0.001	202	0.009	573	0.013	415	0.024
325	23,299	0.826	26	0.006	337	0.012	853	0.058	198	0.029	671	0.102	5808	0.260	11,443	0.252	3067	0.177
336	598	0.021	26	0.006	1068	0.039	25	0.002	210	0.031	114	0.017	1419	0.064	748	0.016	324	0.019
352	8571	0.304	1787	0.399	50,966	1.856	10,631	0.729	8835	1.299	2975	0.453	23,951	1.074	21,634	0.477	18,736	1.079
361	35,679	1.264	9197	2.051	44,529	1.621	22,010	1.508	8490	1.249	12,063	1.839	23,201	1.040	27,088	0.597	16,647	0.959
432	1	0.000	0	0.000	9	0.000	15	0.001	3	0.000	0	0.000	10	0.000	13	0.000	3	0.000
446	15	0.001	0	0.000	0	0.000	0	0.000	0	0.000	0	0.000	1	0.000	13	0.000	1	0.000
472	0	0.000	1	0.000	0	0.000	0	0.000	0	0.000	0	0.000	1	0.000	0	0.000	0	0.000
511	7455	0.264	591	0.132	8208	0.299	2458	0.168	1575	0.232	1786	0.272	8028	0.360	12,295	0.271	5102	0.294
532	7168	0.254	869	0.194	6763	0.246	5422	0.372	2069	0.304	1025	0.156	6277	0.281	8631	0.190	5559	0.320
615	0	0.000	1	0.000	1	0.000	0	0.000	0	0.000	0	0.000	0	0.000	0	0.000	0	0.000
668	18	0.001	4	0.001	5	0.000	10	0.001	0	0.000	6	0.001	7	0.000	26	0.001	0	0.000
671	1538	0.054	0	0.000	5689	0.207	903	0.062	0	0.000	0	0.000	6	0.000	18,026	0.397	961	0.055
271289	577,000	20.44	26,789	5.97	324,914	11.83	340,360	23.33	145,978	21.47	145,853	22.23	556,686	24.96	1,056,773	23.30	417,572	24.06
331337	1065	0.038	52	0.012	1275	0.046	143	0.010	188	0.028	274	0.042	898	0.040	1253	0.028	788	0.045
379381	564	0.020	22	0.005	254	0.009	36	0.002	64	0.009	86	0.013	126	0.006	512	0.011	159	0.009
409412	14,333	0.508	3	0.001	1468	0.053	569	0.039	739	0.109	216	0.033	1460	0.065	1009	0.022	486	0.028
502503	5849	0.207	324	0.072	12,354	0.450	1636	0.112	1851	0.272	2179	0.332	6266	0.281	6011	0.133	3799	0.219
505507	3540	0.125	506	0.113	4924	0.179	4210	0.289	614	0.090	22	0.003	2567	0.115	17,251	0.380	3506	0.202
570577580584585	125,332	4.441	13,485	3.007	114,327	4.163	67,749	4.643	26,267	3.863	33,172	5.056	87,630	3.929	217,152	4.789	77,687	4.475
600601	35,958	1.274	1254	0.280	32,702	1.191	16,264	1.115	3823	0.562	8021	1.223	29,909	1.341	42,827	0.944	17,065	0.983
Total slaughtered	2,822,288		448,412		2,746,407		1,459,135		679,914		656,049		2,230,130		4,534,853		1,735,829	

**Table 2 animals-07-00094-t002:** Prevalence (number and %) of selected slaughter recording codes in sows slaughtered at the three largest sow slaughterhouses (S10–S12) in Denmark in 2012.

Code/Category	S10		S11		S12	
	No.	%	No.	%	No.	%
120	5	0.052	583	0.263	103	0.101
131	7	0.073	500	0.226	4	0.004
132	10	0.104	396	0.179	143	0.140
141	24	0.250	521	0.235	108	0.106
222	4	0.042	300	0.135	781	0.766
230	0	0.000	148	0.067	31	0.030
250	0	0.000	0	0.000	2	0.002
258	3	0.031	41	0.018	11	0.011
325	2	0.021	4	0.002	124	0.122
336	1	0.010	27	0.012	22	0.022
352	112	1.168	4754	2.145	2999	2.940
361	21	0.219	160	0.072	84	0.082
432	14	0.146	1113	0.502	259	0.254
446	0	0.000	8	0.004	1	0.001
472	345	3.596	12,860	5.802	1721	1.687
511	95	0.990	4384	1.978	1815	1.779
532	18	0.188	1361	0.614	399	0.391
615	178	1.856	905	0.408	700	0.686
668	29	0.302	6180	2.788	219	0.215
671	0	0.000	0	0.000	2	0.002
271289	818	8.527	40,100	18.092	23,947	23.477
331337	1	0.010	82	0.037	18	0.018
379381	2	0.021	96	0.043	39	0.038
409412	0	0.000	71	0.032	125	0.123
502503	45	0.469	3076	1.388	1259	1.234
505507	39	0.407	218	0.098	70	0.069
570577580584585	649	6.765	25,012	11.285	11,383	11.160
600601	17	0.177	153	0.069	167	0.164
Total slaughtered	9593		221,645		102,002	

**Table 3 animals-07-00094-t003:** Prevalence of selected slaughter recording codes or categories in 212,826 cattle <18 months of age slaughtered at the eight largest cattle slaughterhouses (C1–C8) in 2012.

Code/Category	C1		C2		C3		C4		C5		C6		C7		C8	
	No.	%	No.	%	No.	%	No.	%	No.	%	No.	%	No.	%	No.	%
120	3	0.0106	3	0.0055	7	0.0162	15	0.0306	4	0.0458	0	0	1	0.0294	2	0.0099
131	1	0.0035	5	0.0092	8	0.0185	1	0.002	0	0	1	0.0168	1	0.0294	0	0
141	19	0.0674	71	0.1311	17	0.0393	82	0.1673	1	0.0115	6	0.1006	0	0	9	0.0448
230	5	0.0177	10	0.0185	18	0.0416	26	0.053	2	0.0229	2	0.0335	0	0	2	0.0099
291	0	0	11	0.0203	5	0.0116	41	0.0836	0	0	0	0	0	0	0	0
325	12	0.0426	9	0.0166	35	0.0809	31	0.0632	0	0	3	0.0503	0	0	27	0.1343
336	0	0	0	0	1	0.0023	2	0.0041	0	0	0	0	0	0	0	0
361	3	0.0106	0	0	0	0	7	0.0143	3	0.0344	0	0	0	0	2	0.0099
412	11	0.039	83	0.1533	54	0.1247	94	0.1918	83	0.951	68	1.14	3	0.0882	6	0.0298
432	0	0	0	0	0	0	3	0.0061	0	0	0	0	0	0	0	0
446	0	0	0	0	0	0	0	0	0	0	0	0	0	0	0	0
509	13	0.0461	2	0.0037	3	0.0069	36	0.0734	0	0	0	0	1	0.0294	0	0
511	6	0.0213	20	0.0369	22	0.0508	30	0.0612	1	0.0115	1	0.0168	0	0	5	0.0249
532	94	0.3337	192	0.3546	311	0.7184	156	0.3183	44	0.5041	6	0.1006	6	0.1764	113	0.5619
572	0	0	1	0.0018	0	0	48	0.0979	0	0	0	0	0	0	0	0
600	6	0.0213	1	0.0018	2	0.0046	12	0.0245	0	0	0	0	0	0	0	0
603	1	0.0035	1	0.0018	0	0	6	0.0122	1	0.0115	0	0	3	0.0882	0	0
668	0	0	0	0	0	0	0	0	0	0	0	0	0	0	0	0
807	0	0	0	0	0	0	0	0	0	0	0	0	0	0	0	0
222223352	344	1.221	2230	4.1186	3758	8.6812	1420	2.8971	158	1.8103	129	2.1626	66	1.94	414	2.0587
271289	726	2.5769	4597	8.4903	5956	13.7587	6377	13.0103	104	1.1916	287	4.8114	103	3.0276	597	2.9687
374375377379381	2788	9.896	8381	15.4791	3418	7.8958	6282	12.8165	1313	15.0435	551	9.2372	220	6.4668	2876	14.3013
472476	0	0	2	0.0037	0	0	1	0.002	0	0	0	0	0	0	0	0
502503	12	0.0426	11	0.0203	23	0.0531	15	0.0306	8	0.0917	3	0.0503	3	0.0882	3	0.0149
505507	50	0.1775	133	0.2456	222	0.5128	149	0.304	20	0.2291	21	0.3521	1	0.0294	34	0.1691
570577580584585	161	0.5715	366	0.676	401	0.9263	539	1.0997	76	0.8708	22	0.3688	7	0.2058	218	1.084
602604	192	0.6815	571	1.0546	846	1.9543	331	0.6753	57	0.6531	138	2.3135	18	0.5291	282	1.4023
631641	0	0	158	0.2918	10	0.0231	18	0.0367	15	0.1719	0	0	0	0	0	0
Total slaughtered	28,173		54,144		43,289		49,015		8728		5965		3402		20,110	

**Table 4 animals-07-00094-t004:** Prevalence of selected slaughter recording codes or combinations (“code”) in 248,580 cattle ≥18 months of age slaughtered at the eight largest cattle slaughterhouses (C1–C8) in 2012.

Code	C1		C2		C3		C4		C5		C6		C7		C8	
	No.	%	No.	%	No.	%	No.	%	No.	%	No.	%	No.	%	No.	%
120	18	0.0486	58	0.115	30	0.0747	158	0.2694	10	0.0661	13	0.0964	12	0.1211	11	0.0462
131	20	0.054	67	0.1328	30	0.0747	79	0.1347	0	0	11	0.0816	9	0.0908	18	0.0756
141	87	0.2351	125	0.2478	87	0.2167	233	0.3973	4	0.0265	39	0.2893	42	0.4237	44	0.1847
230	64	0.173	77	0.1526	91	0.2266	147	0.2507	2	0.0132	15	0.1113	26	0.2623	40	0.168
291	2	0.0054	27	0.0535	10	0.0249	633	1.0794	1	0.0066	0	0	0	0	2	0.0084
325	44	0.1189	60	0.1189	133	0.3312	293	0.4996	2	0.0132	32	0.2374	3	0.0303	42	0.1764
336	0	0	0	0	4	0.01	1	0.0017	0	0	1	0.0074	0	0	0	0
361	2	0.0054	0	0	1	0.0025	2	0.0034	1	0.0066	1	0.0074	0	0	0	0
412	65	0.1757	240	0.4757	162	0.4035	193	0.3291	329	2.1756	351	2.6037	34	0.343	23	0.0966
432	12	0.0324	37	0.0733	19	0.0473	23	0.0392	4	0.0265	11	0.0816	4	0.0404	3	0.0126
446	1	0.0027	1	0.002	0	0	5	0.0085	0	0	0	0	0	0	0	0
509	102	0.2757	10	0.0198	1	0.0025	450	0.7674	1	0.0066	0	0	0	0	0	0
511	30	0.0811	87	0.1724	59	0.1469	168	0.2865	3	0.0198	17	0.1261	21	0.2119	24	0.1008
532	93	0.2513	142	0.2815	263	0.655	304	0.5184	63	0.4166	36	0.267	17	0.1715	138	0.5794
572	0	0	4	0.0079	0	0	875	1.4921	0	0	0	0	0	0	1	0.0042
600	6	0.0162	1	0.002	5	0.0125	76	0.1296	1	0.0066	0	0	0	0	0	0
603	6	0.0162	8	0.0159	7	0.0174	69	0.1177	1	0.0066	1	0.0074	9	0.0908	0	0
668	3	0.0081	2	0.004	3	0.0075	15	0.0256	0	0	0	0	0	0	1	0.0042
807	0	0	14	0.0277	15	0.0374	36	0.0614	0	0	11	0.0816	1	0.0101	0	0
222223352	617	1.6674	3721	7.3752	3249	8.092	3049	5.1993	418	2.7642	543	4.0279	560	5.6497	1091	4.581
271289	419	1.1323	2773	5.4962	1933	4.8143	3454	5.89	86	0.5687	485	3.5977	202	2.0379	486	2.0406
374375377379381	2539	6.8616	8480	16.8077	3226	8.0347	7403	12.6241	2414	15.9635	1341	9.9473	664	6.699	3775	15.8507
472476	218	0.5891	183	0.3627	10	0.0249	148	0.2524	2	0.0132	10	0.0742	2	0.0202	1	0.0042
502503	57	0.154	34	0.0674	57	0.142	60	0.1023	21	0.1389	24	0.178	8	0.0807	17	0.0714
505507	314	0.8486	758	1.5024	968	2.4109	1048	1.7871	136	0.8994	147	1.0904	109	1.0997	282	1.1841
570577580584585	578	1.562	988	1.9583	1169	2.9115	2302	3.9255	201	1.3292	235	1.7432	144	1.4528	602	2.5277
602604	1527	4.1267	3551	7.0382	4308	10.7295	3372	5.7501	297	1.964	1323	9.8138	377	3.8035	1902	7.9862
631641	0	0	10	0.0198	0	0	10	0.0171	1	0.0066	0	0	0	0	0	0
Total slaughtered	37,003		50,453		40,151		58,642		15,122		13,481		9912		23,816	

**Table 5 animals-07-00094-t005:** Selected codes resulting in lack of variance partition estimates due to no model fit (too few positive observations), poor model fit and acceptable model fit for data on pigs and sows.

Group	Model Fit	Codes & Code Combinations
Pigs	No model fit	432, 446, 451, 472, 572, 615, 634
	Poor model fit	230, 250, 258, 668
	Acceptable model fit	120, 131, 132, 141, 222, 325, 336, 352, 361, 511, 532, 671, 271289, 331337, 379381, 409412, 502503, 505507, 570577580584585, 600601
Sows	No model fit	250, 336, 446, 451, 572, 634, 671
	Poor model fit	258, 361, 331337
	Acceptable model fit	120, 131, 132, 141, 222, 230, 325, 352, 432, 472, 511, 532, 615, 668, 271289, 379381, 409412, 502503, 505507, 570577580584585, 600601

**Table 6 animals-07-00094-t006:** Selected codes resulting in lack of estimates due to no model fit (too few positive observations), poor model fit and acceptable model fit for data on cattle.

Animal Group	Model Fit	Codes & Code Combinations
Calves	No model fit	120, 131, 291, 336, 361, 432, 446, 509, 572, 600, 603, 668, 807, 472476
	Poor model fit	230, 325, 511, 502503, 505507, 631641
	Acceptable model fit	141, 412, 532, 271289, 222223352, 374375377379381, 570577580584585, 602604
Cows	No model fit	336, 361, 446, 668, 631641
	Poor model fit	120, 131, 141, 230, 432, 511, 600, 603, 807, 472476, 502503
	Acceptable model fit	291, 325, 412, 509, 532, 572, 271289, 222223352, 374375377379381, 570577580584585, 505507, 602604

**Table 7 animals-07-00094-t007:** Summary of random effect coefficient estimates (on the logit scale) modelled for individual meat inspection codes or categories of codes.

Animal Group	Abattoir Effect	Individual or Category	Intervals ^1^	Number of Codes	Codes
Pigs	No	None	NA	0	
	Yes	12 individual	<|1|	4	120; 361; 511; 532
			|1|–|2|	5	131; 132; 141; 336; 352
			>|2|	3	222; 325; 671
		19 codes in 8 categories	<|1|	11 (4)	331337; 502503; 600601; 570577580584585
			|1|–|2|	4 (2)	271289; 379381
			>|2|	4 (2)	409412; 505507
Sows	No	2 individual	NA	2	132; 230
		4 codes in 2 categories	NA	4 (2)	379381; 600601
	Yes	12 individual	<|1|	8	120; 141; 352; 432; 472; 511; 532; 615
			|1|–|2|	2	222; 668
			>|2|	2	131; 325
		13 codes in 5 categories	<|1|	9 (3)	271289; 502503; 570577580584585
			|1|–|2|	4 (2)	409412; 505507
			>|2|	0	
Cattle < 18 months	No	None	NA	0	
	Yes	4 individual	<|1|	2	141; 532
			|1|–|2|	1	412
			>|2|	0	
		17 codes in 5 categories	<|1|	12 (3)	374375377379381; 570577580584585; 602604
			|1|–|2|	5 (2)	222223352; 271289
			>|2|	0	
Cattle ≥ 18 months	No	1 individual		1	532
	Yes	7 individual	<|1|	0	
			|1|–|2|	2	325; 412
			>|2|	3	291; 509; 572
		19 codes in 6 categories	<|1|	17 (5)	222223352; 505507; 374375377379381; 570577580584585; 602604
			|1|–|2|	2 (1)	271289
			>|2|	0	

^1^ Intervals are absolute values of the coefficients on the logit scale, e.g., the absolute value of −1.2 is 1.2 and it will be in the interval |1|–|2|. Codes in intervals <|1| indicate that the codes might be useful if they accurately predict animal welfare conditions; interval |1|–|2| indicate that the between slaughterhouse differences are deemed so high that it should be considered if application of correction factors will be appropriate; and >|2| indicates major differences between slaughterhouses and application of correction factors is deemed inappropriate. NA: Not applicable as there was no random effect.

**Table 8 animals-07-00094-t008:** Meat inspection codes deemed potentially useful for welfare related purposes given that they are accurate, while the abattoir effect is significant for most but still within a relatively small range.

Swine Code	Cattle Code	Description	Useful in
120		Circulatory system disturbances (poor bleeding); anaemia; dropsy; oedema	pigs; sows
132		Skinny	sows
141		Pyemia; septicaemia; pyemic lung abscesses; splenitis-septicaemia; nephritis-septicaemia;	sows
	141	Pyemia; septicaemia; pyemic lung abscesses; splenitis-septicaemia; nephritis-septicaemia; pyemic hepatic abscesses	calves
	222223352	Chronic pericarditis; Traumatic reticulitis-pericarditis; Chronic peritonitis; peritoneal abscess incl. subphrenic abscesses	cows
230		Endocarditis (acute or healed)	sows
271289		Chronic pneumonia or pleuritis; aeronic abscesses; serositis	sows
331337		Rectal prolapse; rectal stricture	pigs
352		Chronic peritonitis; peritoneal abscess; discoloured peritoneum (from splenic torsion)	sows
361		Hernia (umbilical; inguinal)	pigs
	374375377379381	Fatty liver; acute, subacute, chronic hepatic abscesses and non-pyemic abscesses; chronic hepatitis with necrosis; chronic parasitic hepatitis; liver cirrhosis; jaundice	calves; cows
379381		Chronic hepatitis; hepatic necrosis; jaundice	sows
432		Chronic metritis; retained placenta; incomplete parturition; uterine prolapse	sows
472		Chronic mastitis	sows
502503		Old fracture; infected fracture; open fracture >6 h old	pigs; sows
511		Acute, chronic, local, healed osteomyelitis; abscesses following wound	pigs; sows
	505507	Tail fracture; rib fracture, healed	cows
532	532	Chronic arthritis; arthrosis	All
570577580 584585		Abscesses in front, mid or rear part; in the leg or toe; in the head; blood ear	pigs; sows
	570577580584585	Abscesses in front, mid or rear part; in the leg or toe; in the head; tongue incl. actinomycosis	calves; cows
600601		Tail-bite, local; tail-bite incl. Infection	pigs; sows
	602604	Hock, hip; chest, thigh, pinbone, ischial abrasions	calves; cows
615		Shoulder wounds	sows

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
