# Peer review of "Selection of Meat Inspection Data for an Animal Welfare Index in Cattle and Pigs in Denmark"

_animals, 2017, doi:10.3390/ani7120094_

Round 1

Reviewer 1 Report

The authors assessed the potential for using data from the systematic meat inspection to contribute to animal welfare indices.

The objectives of the present study were to provide statistical assessment of meat inspection data to a) select codes of relevance to an animal welfare index based on prevalence and welfare impact; b) assess the contribution of each slaughterhouse on the variation in prevalence of each relevant meat inspection variable; and c) provide estimates of a correction factor for each slaughterhouse for each of the relevant meat inspection code.

Some potentially useful codes (parameters) for a welfare index were identified. Universal correction factors could not be found but approaches for potential corrections were suggested and discussed.

The study and the approach are interesting and the manuscript is well written. Some methodological aspects need to be clarified. Future questions to target/address before the parameters could be included in an animal welfare index should be added in the discussion. For instance, it should be checked if results are stable/repeatable over at least two years. At least for the recommended codes this would be interesting.    

Further comments:

Simple summary:

Ln 16: “lack of subjectivity” => because a fracture should be easy to diagnose?

Ln 20: “Furthermore, other factors are important to also monitor welfare of both cattle and pigs” What kind of factors (examples)?

Abstract

Ln 32-33: The absolute accuracy of each code/combination could not be assessed, only the relative variation between farms and abattoirs. => point out /hint at next steps that would be necessary for including the assessment of these codes (parameters) at slaughterhouse Level into animal welfare indices

 Material and Methods

Who did the meat inspections? Is it vets? Other trained staff? Do all slaughter houses have their own protocols? Were some of the slaughter houses linked (e.g., via a company)?

From how many farms per species do the data come?

How many pig abattoirs were included? For cattle, it said it was 8 slaughter houses (ln 164); in Table 1 descriptive data for 9 pig abattoirs are shown. How were pig slaughter houses selected? For cattle it was indicated that slaughterhouses with less than 10,000 cattle slaughtered in 2012 were not included

Ln 86: “codes were excluded if they had a low prevalence combined with a low impact on welfare” => give examples, or refer to the appendix?

Ln89: “all codes associated to included liver conditions” strange wording

Ln 96: “for each code” not “each codes”

Results see comments regarding Tables & Figure

Discussion

Please discuss more critically steps that need to be considered before implementing this approach in an welfare index. E.g., test repeatability /consistency over years. It would be also interesting to know who should later analyse the data and if the data are then reported back to farms or if some thresholds or limits should be suggested with legal consequences for the farmers. Then it would be worth validating if prevalences found at the slaughterhouses correspond to problems that can be assessed on-farm.

Ln 260: please mention in the discussion what the codes stand for ; e.g., “skinny” otherwise one always has to check the appendix

Ln 263: please mention what the code stands for (Chronic arthritis, osteoarthritis)

Table 1

check the formatting, heading is shifted, should fit on a page (if the journal does not edit it), explain the abbreviation “No.”: Number of what? Is it the number of affected animals?

Table 2

explain the abbreviation “No.”

Table 3

check the formatting, heading is shifted, should fit on a page (if the journal does not edit it), explain the abbreviation “No.”; in Table 1 & 3 it says “Total”, here it says “total slaughtered)

Table 7

Explain “NA” in the “Intervals” column;

Table 8

Ln 244: Check wording: “deemed potentially useful due for welfare”

Figure 1:

Figure text indicates colours in the figure (“for each code in pigs (blue) and sows (pink)”), but it is grey and black

Author Response

Author (AU) responses to Reviewer 2 comments

The authors assessed the potential for using data from the systematic meat inspection to contribute to animal welfare indices.

The objectives of the present study were to provide statistical assessment of meat inspection data to a) select codes of relevance to an animal welfare index based on prevalence and welfare impact; b) assess the contribution of each slaughterhouse on the variation in prevalence of each relevant meat inspection variable; and c) provide estimates of a correction factor for each slaughterhouse for each of the relevant meat inspection code.

Some potentially useful codes (parameters) for a welfare index were identified. Universal correction factors could not be found but approaches for potential corrections were suggested and discussed.

The study and the approach are interesting and the manuscript is well written. Some methodological aspects need to be clarified. Future questions to target/address before the parameters could be included in an animal welfare index should be added in the discussion. For instance, it should be checked if results are stable/repeatable over at least two years. At least for the recommended codes this would be interesting

AU: We have added additional information on the methodology according to all reviewers’ specific comments. With a 10-day deadline it was not really feasible to repeat the study, but we have referenced one study demonstrating that time is a challenge.

Further comments:

Simple summary:

Ln 16“lack of subjectivity” => because a fracture should be easy to diagnose?

AU: Corrected as suggested

Ln 20: “Furthermore, other factors are important to also monitor welfare of both cattle and pigs”What kind of factors (examples)?

AU: Added as suggested

Abstract

Ln 32-33: The absolute accuracy of each code/combination could not be assessed, only the relative variation between farms and abattoirs. => point out /hint at next steps that would be necessary for including the assessment of these codes (parameters) at slaughterhouse Level into animal welfare indices

AU: There is no more space in the Abstract to go into this aspect, but the steps that may follow has been hinted at in the Discussion.

Material and Methods

Who did the meat inspections? Is it vets? Other trained staff? Do all slaughter houses have their own protocols? Were some of the slaughter houses linked (e.g., via a company)?

AU: This information was added 

From how many farms per species do the data come?

AU: This information was added 

How many pig abattoirs were included? For cattle, it said it was 8 slaughter houses (ln 164); in Table 1 descriptive data for 9 pig abattoirs are shown. How were pig slaughter houses selected? For cattle it was indicated that slaughterhouses with less than 10,000 cattle slaughtered in 2012 were not included

AU: This information was added

Ln 86: “codes were excluded if they had a low prevalence combined with a low impact on welfare” => give examples, or refer to the appendix?

AU: Examples were given

Ln89: “all codes associated to included liver conditions” strange wording

AU: Rephrased

Ln 96: “for each code” not “each codes”

AU: Corrected as suggested

Results see comments regarding Tables & Figure

Discussion

Please discuss more critically steps that need to be considered before implementing this approach in an welfare index. E.g., test repeatability /consistency over years. It would be also interesting to know who should later analyse the data and if the data are then reported back to farms or if some thresholds or limits should be suggested with legal consequences for the farmers. Then it would be worth validating if prevalences found at the slaughterhouses correspond to problems that can be assessed on-farm.

AU: A number of aspects have been discussed in greater detail as suggested by this reviewer and the other reviewers

Ln 260: please mention in the discussion what the codes stand for ; e.g., “skinny” otherwise one always has to check the appendix

AU: Done as suggested

Ln 263: please mention what the code stands for (Chronic arthritis, osteoarthritis)

AU: Information added

Table 1. check the formatting, heading is shifted, should fit on a page (if the journal does not edit it), explain the abbreviation “No.”: Number of what? Is it the number of affected animals?

AU: Done

Table 2. explain the abbreviation “No.”

AU: It is total slaughtered. Updated in all tables. Formatting was also updated (in this and other tables).

Table 3. check the formatting, heading is shifted, should fit on a page (if the journal does not edit it), explain the abbreviation “No.”; in Table 1 & 3 it says “Total”, here it says “total slaughtered)

AU: It is total slaughtered. Updated in all tables. Formatting was also updated (in this and other tables).

Table 7 Explain “NA” in the “Intervals” column;

AU: “NA” explained 

Table 8 Ln 244: Check wording: “deemed potentially useful due for welfare”

AU: Rephrased

Figure 1: Figure text indicates colours in the figure (“for each code in pigs (blue) and sows (pink)”), but it is grey and black

AU: Now updated to blue and pink

Reviewer 2 Report

Line No

Original Text

Comment

21

Abstract: National welfare   indices of cattle and pigs are constructed in Denmark

Define WI – “welfare indices” are an attempt to convert a complicated   valued concept to a number which allows communication of that valued concept   to politicians and the public (1)

47

there was also a desire to include register data

48 whenever possible.

Define WI – RD is numerical information collected as prat of the   organoleptic inspection of live animals and carcass post mortem during the   process of slaughter.  This data is   highly standardized where specific observations are recorded as a “code” in   the national database. This database is sufficiently rigorous to trace   individual animals to farm of origin.

65

The assessments were done separately for cattle aged < 18 months   (hereafter denoted 'calf'),

The use of “calf” is vague to north American readers. For census   purposes a “Calf” is less than 300kg on October first.  It is unclear from the paper if < 18   months includes Veal calves less than 300kg and veal calves less than   600kg. 

The authors use “market hogs” to differentiate from sows why not use “market   cattle” to differentiate from cows. 

Not a hill to die on for certain.  Does the veal question need to be answered?    

68

Meat inspection data for 2012

30 words to brag about the data capture may be indicated. “Coding of observations   at slaughter inspection are recorded electronically at the carcass inspection   station, verified by government veterinarians and uploaded daily (weekly) to   the National database called “well for the holding of Meat information”.  That data was made available to the authors…….  

73

This was referred to as a “batch”,

Use of English [trivia].  Batch   in food processing in North America consistently applies to a temporal   production of X.  We have plants that   do continual production of ground meat and places that do batch processing. Recall   is done on the batch identity.  I think   the word you are looking for is better described by cohort. However; cohort in   epidemiology normally is also usually temporally restricted such as all the   male children born in Denmark in the calendar year of 2010 would be the 2010   male cohort.

Truth is, we don’t have a good word in English to describe all the   pigs from Farm X that went to slaughterhouse Y during the calendar year.  This complex cluster of conditions to   describe a population is rare in the literature, sometimes called a value   chain in method of production labelling circles.  As defined in the paper it was clear, just a   novel use of the word “batch”

80

the purpose of the study, which was to assess

81 changes in   on-farm welfare of cattle and pigs,

This specific purpose was not articulated in the Abstract.  The abstract does not communicate the   effort to parse out what findings in the slaughter data are indicative of   animal welfare concerns ORIGINATING ON THE FARM.

90

in cattle were combined (374, 375, 377, 379, 381 to 374375377379381,

This is not an attractive way to communicate (personal preference).  What I understood is that codes which were   judged to be equivocations as far as animal welfare was concerned were collapsed   into a single category.  Collapse is a   better descriptor of data management in the manner described.

126-129

We note that a subset of this data 126 has already been presented to   illustrate the statistical methodology developed to analyse the data 127   [13], but here we consider the welfare implications of the analyses rather   than the statistical methods 128 themselves, and also widen the scope to   include both pigs and cattle.

Frankly Section 2.2 was beyond my statistical incantation powers. The   editor should give some concern that Reference 13

Denwood, M.J., Houe, H., Forkman, B.,   Nielsen, S.S. Random effect selection in generalised linear models: A   practical application to slaughterhouse surveillance data in Denmark. In:   Thulke, H.-H. and Verheyen K. (eds). Proceedings of the Society for   Veterinary Epidemiology and Preventive Medicine annual meeting 382 held in   Ghent, SVEPM, 2015, pp. 135-145.

Is not from an Elsevier level peer reviewed statistical Journal so I   fail to criticize this methodology (Sec 2.2) based on faith, Lord help my   unbelief. KJV Mark 9:25

144

Bullet Points

Can we convert this to written?    I hate bullet form in scientific literature. I work for the government   and the primary use of bullet format is so the completely ignorant can   convince themselves they understand issues beyond their mental capacity.  Power-point has taken over as the sacred   format of communicating in the government service.  Max Weber should be rolling in his   grave.  Can we eliminate the code   numbers that are clear in the tables?

181-183

Tables

Somewhere in the methods can you note that hog killing plants were   specialized in that sow plants sows and market hog plants kill markets

Whereas

All beef plants killed both market cattle and cows.  I had to flip through the tables a couple of   times before I figured that out myself.

185, 187

Table 3 & Table 4

The beef table header that sums the total of class killed.  This is not in the Pig tables.  From an Ascetic point of view the Total   Slaughtered line of each table is begging for addition of a cell at the far   right, horizontal sum of the subtotals.

For example if I am reading the tables correctly, in Table 4, Cows   there were 248580 individuals in this class slaughtered in the calendar year and   497160 defect codes generated.  I am   uncertain what this means if anything in your approach to the data analysis

197

the variance effect due to abattoirs was 0.29, the farm effect was   0.38 and the residual 0.15.

(Demonstrating my ignorance) Does this statement mean that 38% of the   variance in the data can be attributed to farm associated codes?  If this is true, I would add it after this   sentence to compensate for the statistically impaired reader.

214

Figure 1

Circulated draft was not in colour but I figured it out anyway.  The final e-file should definitely be in   colour.

265

although some pairs of codes (for example Codes 336 and 120 in pigs)

These code numbers mean nothing to the reader “although some pairs of   codes; for example Codes 336 (gastric ulcer) and 120 (anemia) in pigs)…Now   this statement is intuitively reasonable and transmits meaning to the reader.

287

When the differences are smaller, then use of a correction factor to   ‘even out' small variations between abattoirs may be useful to allow a more   robust comparison of observed farm prevalences.

Not completely comfortable with a suggestion that you can bugger with   your raw data to better get the answer you want. Perhaps I have misread the statement.

296

Differences in accuracy of recording practices are likely to be the   main cause

Consistency in veterinary disposition is widely accepted as a problem   in organoleptic food safety programs.

Try to find a single study where there was a critical review of  inter-observer agreement veterinary   disposition meat.   

Overall:  There is an elephant in the room that needs to be addressed in the discussion part of this paper.  This is an enormous project of data-dredging and for scientific integrity the authors have to give some estimation of the risk of identifying associations that make no biological sense or that appear due to the expected false positive rate. 

 These guys are pretty savvy with understanding data.  I would be really impressed if in the methods they described there data set a little better and calculated the Power(2) prior to hunting for associations.  This is a significant modification to the paper but I think it would add a lot because the statistics used are pretty foreign to a mainstream reader of this Journal.

1.            Hansen HK, Mühlen-Schulte A. The power of numbers in global governance. Journal of International Relations and Development. 2012;15(4):455-65.

2.            Nuzzo RL. Statistical Power. PM&R. 2016;8(9):907-12.

Author Response

Author (AU) responses to Reviewer 1 comments

Overall:  There is an elephant in the room that needs to be addressed in the discussion part of this paper.  This is an enormous project of data-dredging and for scientific integrity the authors have to give some estimation of the risk of identifying associations that make no biological sense or that appear due to the expected false positive rate.

These guys are pretty savvy with understanding data.  I would be really impressed if in the methods they described there data set a little better and calculated the Power(2) prior to hunting for associations.  This is a significant modification to the paper but I think it would add a lot because the statistics used are pretty foreign to a mainstream reader of this Journal.

1. Hansen HK, Mühlen-Schulte A. The power of numbers in global governance. Journal of International Relations and Development. 2012;15(4):455-65.

2. Nuzzo RL. Statistical Power. PM&R. 2016;8(9):907-12.

AU: In principle, this is not about estimation associations but quantifying effects. The methodology used is reasonably standard. We have clarified these aspects in the Discussion

Line No

Original Text

Comment

AU Response

21

Abstract: Nationalwelfare   indices of   cattle and pigs are constructed in Denmark

Define WI – “welfare indices” are an attempt to   convert a complicated   valued concept to a number which allows   communication of that valued concept   to politicians and the public (1)

We   have not defined the indices in the Abstract due to lack of space.

We   have extracted the purpose and stated this, just following the reference to   the parliament decision.

We   have added a reference to the suggested paper highlighting the point of the reviewer.   This is done in the discussion of the paper

47

there was also a desire to include register data

 whenever   possible.

Define WI – RD is numerical information collected as   prat of the   organoleptic inspection of live animals and carcass post   mortem during the   process of slaughter.  This data is     highly standardized where specific observations are recorded as a “code” in     the national database. This database is sufficiently rigorous to trace     individual animals to farm of origin.

We are not   defining the indices here, as they are beyond the scope of the present work   and not fur us to define here. We have added a point about use of indices to   the discussion. However, we have added references to the follow-up work

65

The assessments were done separately for cattle aged   < 18 months   (hereafter denoted 'calf'),

The use of “calf” is vague to north American   readers. For census   purposes a “Calf” is less than 300kg on October   first.  It is unclear from the paper if < 18   months   includes Veal calves less than 300kg and veal calves less than     600kg. 

The authors use “market hogs” to differentiate from   sows why not use “market   cattle” to differentiate from cows. 

Not a hill to die on for certain.  Does the   veal question need to be answered?    

It was numerically   defined because concepts of “veal”, “beef” and production system includes a   variety of definitions, which may as well be country and producer specific.

68

Meat inspection data for 2012

30 words to brag about the data capture may be   indicated. “Coding of observations   at slaughter inspection are   recorded electronically at the carcass inspection   station, verified by   government veterinarians and uploaded daily (weekly) to   the National   database called “well for the holding of Meat information”.  That   data was made available to the authors…….  

This information   is added.

73

This was referred to as a “batch”,

Use of English [trivia].  Batch   in   food processing in North America consistently applies to a temporal     production of X.  We have plants that   do continual   production of ground meat and places that do batch processing. Recall     is done on the batch identity.  I think   the word you are   looking for is better described by cohort. However; cohort in     epidemiology normally is also usually temporally restricted such as all the     male children born in Denmark in the calendar year of 2010 would be the   2010   male cohort.

Truth is, we don’t have a good word in English to   describe all the   pigs from Farm X that went to slaughterhouse Y during   the calendar year.  This complex cluster of conditions to     describe a population is rare in the literature, sometimes called a value     chain in method of production labelling circles.  As defined   in the paper it was clear, just a   novel use of the word “batch”

We have retained   the word “batch” as suggested while it is defined

80

the purpose of the study, which was to assess   changes in on-farm welfare of cattle and pigs

This specific purpose was not articulated in the   Abstract.  The abstract does not communicate the   effort to   parse out what findings in the slaughter data are indicative of   animal   welfare concerns ORIGINATING ON THE FARM.

There is a   difference between the objective (given by us) and the purpose (given by   decision makers). We have to relate to the purpose given by decision makers   (given in the Introduction) – hence this formulation

90

in cattle were combined (374, 375, 377, 379, 381 to   374375377379381,

This is not an attractive way to communicate   (personal preference).  What I understood is that codes which were     judged to be equivocations as far as animal welfare was concerned were   collapsed   into a single category.  Collapse is a     better descriptor of data management in the manner described.

This suggestion has   been included throughout the text.

126-129

We note that a subset of this data has already been   presented to   illustrate the statistical methodology developed to   analyse the data  [13], but here we   consider the welfare implications of the analyses rather   than the   statistical methods themselves, and also widen the scope to   include   both pigs and cattle.

Frankly Section 2.2 was beyond my statistical   incantation powers. The   editor should give some concern that Reference   13

Denwood, M.J., Houe, H., Forkman, B.,     Nielsen, S.S. Random effect selection in generalised linear models: A     practical application to slaughterhouse surveillance data in Denmark. In:     Thulke, H.-H. and Verheyen K. (eds). Proceedings of the Society for     Veterinary Epidemiology and Preventive Medicine annual meeting 382   held in   Ghent, SVEPM, 2015, pp. 135-145.

Is not from an Elsevier level peer reviewed   statistical Journal so I   fail to criticize this methodology (Sec 2.2)   based on faith, Lord help my   unbelief. KJV Mark 9:25

The other   reviewers seem to have insight into the methodology, which is relatively   straight-forward.

Furthermore, other   relevant comments have been given by the reviewer, so there seems to be a   nice overlap between competences

144

Bullet Points

Can we convert this to written?    I   hate bullet form in scientific literature. I work for the government     and the primary use of bullet format is so the completely ignorant can     convince themselves they understand issues beyond their mental   capacity. Power-point has taken over as the sacred   format of   communicating in the government service.  Max Weber should be   rolling in his   grave.  Can we eliminate the code     numbers that are clear in the tables?

Converted

181-183

Tables

Somewhere in the methods can you note that hog   killing plants were   specialized in that sow plants sows and market hog   plants kill markets

Whereas

All beef plants killed both market cattle and cows.    I had to flip through the tables a couple of   times before I   figured that out myself.

This information has been   clarified in the Materials and Methods

185, 187

Table 3 & Table 4

The beef table header that sums the total of class   killed.  This is not in the Pig tables.  From an Ascetic point   of view the Total   Slaughtered line of each table is begging for   addition of a cell at the far   right, horizontal sum of the subtotals.

For example if I am reading the tables correctly, in   Table 4, Cows   there were 248580 individuals in this class slaughtered   in the calendar year and   497160 defect codes generated.  I am     uncertain what this means if anything in your approach to the data   analysis

No,   this is not the way it should be interpreted.

The   total slaughtered have been added the tables they were not present in, and   the total columns for the cattle for disease codes were removed, as they are   irrelevant.

197

the variance effect due to abattoirs was 0.29, the   farm effect was   0.38 and the residual 0.15.

(Demonstrating my ignorance) Does this statement   mean that 38% of the   variance in the data can be attributed to farm   associated codes?  If this is true, I would add it after this     sentence to compensate for the statistically impaired reader.

No. It cannot be   interpreted like that for binomial data. We have thus not made any changes.

214

Figure 1

Circulated draft was not in colour but I figured it   out anyway.  The final e-file should definitely be in     colour.

Now updated to   blue and pink

265

although some pairs of codes (for example Codes 336   and 120 in pigs)

These code numbers mean nothing to the reader   “although some pairs of   codes; for example Codes 336 (gastric ulcer)   and 120 (anemia) in pigs)…Now   this statement is intuitively reasonable   and transmits meaning to the reader.

Amended as suggested

287

When the differences are smaller, then use of a   correction factor to   ‘even out' small variations between abattoirs may   be useful to allow a more   robust comparison of observed farm   prevalences.

Not completely comfortable with a suggestion that   you can bugger with   your raw data to better get the answer you want.   Perhaps I have misread the statement.

If there are   differences that is not due to biology, then this is one way to do it. This   is also known from other diagnostic tests.

296

Differences in accuracy of recording practices are   likely to be the   main cause

Consistency in veterinary disposition is widely   accepted as a problem   in organoleptic food safety programs.

Try to find a single study where there was a   critical review of  inter-observer agreement veterinary     disposition meat.   

Agreement paper added

Reviewer 3 Report

General comments:

It is a nice paper about the potential of some meat inspection data for animal welfare assessments. It has some nice ideas on how to select codes out of all available data. Unfortunately some important steps are not included at all in the analysis. To really assess the potential of codes to predict animal welfare this should be included. The lack of this last part makes the paper less interesting. In my opinion, it is better to integrate also the next steps (how variable are de data between farms, and is this variance associated with an animal welfare index) in this paper, or to submit as a part I and part II twin submission.

There are also some parts in M&M that need to be clarified and a more strict distinction between M&M and results is needed to avoid repetitions.

My main concern is that it is not investigated if there is selection bias of farm on the level of slaughterhouse. Is the effect of slaughterhouse really due to differences between slaughterhouses or due to preferences of farms for specific slaughterhouses. Some slaughterhouses are known to be less strict than others… Exploring the data should make it possible to provide an idea on this although testing will be difficult unless you have enough farms with known welfare information. If there is a selection bias, you risk that the slaughterhouse-effect is mainly due to the farm-selection. In that case the variance between farms decreases because the between farms is partly absorbed the increasing slaughterhouse-variance. This will result in rejection of the code while it could be an interesting proxy for animal welfare.

Specifiec comments    

72 Why a two-step aggregation of the data. This has potential problems such as over or under-estimation of prevalences due to small numbers of animals in one slaughter date. E.g. two slaughter dates: A: 2 cows, one with code X: prevalence=0.5, B: 18 cows, none with code X=prevalence = 0.

 Two-step aggregation: prevalence=0.25

 One-step aggregation: prevalence=0.05

In this case I think a one-step aggregation is more appropriate!

78 rephrase: [Some irrelevant “commercial codes” (such as information about contamination, missing organs and slaughter line issues) were excluded from the data.] “initially excluded” suggests that you included them again for the final analysis.

90 How was this combination done, please detail the method? If the codes were not exclusive: was there a correction to avoid double coding (eg if not exclusive: 2 cows, one with both code 374 and 375: results in prevalence of 1 or 0.5?)

95 How was this consensus assessed?

97 As far as I know you can’t model aggregated prevalences using glmer although you suggest a two-step aggregation in line 72 to get prevalences. Did you really model prevalances (a proportion) or a binomial response (0/1) or the number of occurences/number of not affected animals (cbind(n_occ,n_animals-n_occ)?

106 Do you think overfitting in a model with more than 200 000 observations will be an issue? You model at most 4 parameters (intercept and 3 random components), this part seems irrelevant to me in this paper. A much more important issue in your paper is the overpowered analysis which can result in significant but irrelevant effects.

108 Am I right that batch is nested both in farm and in slaughterhouse? Did you check if this is in most cases true or is batch sometimes the same as the farm effect (eg a farm that uses always the same slaughterhouse) which results in a batch=farm effect which is nested within slaughterhouse? If this is the case, you cannot exclude that part of the slaughterhouse is actually due to a farm-effect. This is typically not the case in randomized trials, where you try to balance the random effects, but this is an observational study! Did you take this into account or at least comment on this in the discussion?

131 If you exclude the codes due to a significant random slaughterhouse effect you assume that farms go to at least 2 or more slaughterhouses in a balanced way. Did you check this assumption? Eg. if all bad farms only go to slaughterhouse A and all good farms only to slaughterhouse B, there will be almost no farm effect but a huge slaughterhouse effect… If both good and bad farms each go to slaughterhouse A and B there will be a huge farm effect and almost no slaughterhouse effect…

139 in my opinion corrections should always be done on the logit scale!

139 Did you consider a model with slaughterhouse as fixed effect to asses significance of the slaughterhouse effect. You can still nest farm and batch within slaughterhouse as random component. Using such a model you can easily detect the biggest differences and conclude if correction is needed or not.

140 and smaller significant differences means no systematic difference? You talk about corrections, how will you perform than in practice? Or do you assume that differences less than 2 (on logit scale) can be neglected?

Please state your opinion in these differences

Are there inclusion criteria for batch or are all batches included?

147-160: this information is already in M&M. please only refer to the tables

166-176 idem

162-163: this is M&M move this to the appropriate part

177 I miss the number of farms, slaughterhouses and batches and the total number of animals included in this paper,

200 I really don’t like comparing variances of the random effect parts. But how did you test for the differences you state here? I am not yet convinced that 0.36 is bigger than 0.26 if you have a limited number of slaughterhouses, this will probably be borderline? Please also report test-statistics and p-values for stated differences.

Fig 1 do S10-12 exclusively do sows? And the other only pigs? Very strange, this is uncommon in other countries.. Why random order of codes, this makes it difficult to find a specific code

217 Abattoir A71461? = Cx?

Fig 2 Why do you not integrate the calves and cows in one graph, this would be very interesting to see if differences are in the same line for calves and cows, they are slaughtered in the same slaughterhouses which make it more interesting to combine compared to Fig 1 where you did combine them although there were no slaughterhouses doing both types of animals. Same remark on ordering

220 why do you expect an overall systematic difference for all codes the same, seems rather unlikely to me?

254 how big is the group that exclusively send animals to only one slaughterhouse? Did you consider to exclude those farms in order to have a better estimation of the slaughterhouse-effect

309-310 which is probably the case. Did you test how balanced division over slaughterhouses within the same farm was. If a large amount of farms go with a very small proportion of animals to one specific slaughterhouse this can be an indication;…

I miss the interpretation of farm-effect itself. If you have a very low slaughterhouse-effect it is a promising code in your opinion, but if also the farm-effect is very low, the added value for a WQ-index will be very limited in my opinion! 

Author Response

Author (AU) responses to Reviewer 3 comments

General comments:

It is a nice paper about the potential of some meat inspection data for animal welfare assessments. It has some nice ideas on how to select codes out of all available data. Unfortunately some important steps are not included at all in the analysis. To really assess the potential of codes to predict animal welfare this should be included. The lack of this last part makes the paper less interesting. In my opinion, it is better to integrate also the next steps (how variable are de data between farms, and is this variance associated with an animal welfare index) in this paper, or to submit as a part I and part II twin submission.

There are also some parts in M&M that need to be clarified and a more strict distinction between M&M and results is needed to avoid repetitions.

My main concern is that it is not investigated if there is selection bias of farm on the level of slaughterhouse. Is the effect of slaughterhouse really due to differences between slaughterhouses or due to preferences of farms for specific slaughterhouses. Some slaughterhouses are known to be less strict than others… Exploring the data should make it possible to provide an idea on this although testing will be difficult unless you have enough farms with known welfare information. If there is a selection bias, you risk that the slaughterhouse-effect is mainly due to the farm-selection. In that case the variance between farms decreases because the between farms is partly absorbed the increasing slaughterhouse-variance. This will result in rejection of the code while it could be an interesting proxy for animal welfare.

AU: We have added information about additional steps and provided further clarification about the materials and methods as commented in the specific comments from all reviewers

Specific comments    

72 Why a two-step aggregation of the data. This has potential problems such as over or under-estimation of prevalences due to small numbers of animals in one slaughter date. E.g. two slaughter dates: A: 2 cows, one with code X: prevalence=0.5, B: 18 cows, none with code X=prevalence = 0.

 Two-step aggregation: prevalence=0.25

 One-step aggregation: prevalence=0.05

In this case I think a one-step aggregation is more appropriate!

AU: You are right. We have removed this sentence because it causes confusion

78 rephrase: [Some irrelevant “commercial codes” (such as information about contamination, missing organs and slaughter line issues) were excluded from the data.] “initially excluded” suggests that you included them again for the final analysis.

AU: Rephrased as suggested 

90 How was this combination done, please detail the method? If the codes were not exclusive: was there a correction to avoid double coding (eg if not exclusive: 2 cows, one with both code 374 and 375: results in prevalence of 1 or 0.5?)

AU: This was clarified also to address the comments from reviewer 1

95 How was this consensus assessed?

AU: The consensus itself wasn’t addressed. The sentence has been rephrased

97 As far as I know you can’t model aggregated prevalences using glmer although you suggest a two-step aggregation in line 72 to get prevalences. Did you really model prevalances (a proportion) or a binomial response (0/1) or the number of occurences/number of not affected animals (cbind(n_occ,n_animals-n_occ)?

AU: This is true. We have deleted the first sentence as it is really redundant and the details are given below

106 Do you think overfitting in a model with more than 200 000 observations will be an issue? You model at most 4 parameters (intercept and 3 random components), this part seems irrelevant to me in this paper. A much more important issue in your paper is the overpowered analysis which can result in significant but irrelevant effects.

AU: We have removed reference to “overfitting”. With regards to significant but irrelevant effects, then this is not an issue, as we present no p-values … so therefore have no significant effects at all, and as we are not doing a hypothesis test there is no concept of power either.  We report confidence intervals which the reader can interpret as being biologically important or not

108 Am I right that batch is nested both in farm and in slaughterhouse? Did you check if this is in most cases true or is batch sometimes the same as the farm effect (eg a farm that uses always the same slaughterhouse) which results in a batch=farm effect which is nested within slaughterhouse? If this is the case, you cannot exclude that part of the slaughterhouse is actually due to a farm-effect. This is typically not the case in randomized trials, where you try to balance the random effects, but this is an observational study! Did you take this into account or at least comment on this in the discussion?

AU: There are some farms that only use 1 slaughterhouse. We don’t think it is a problem for slaughterhouse effects (slaughterhouses always have more than 1 farm) but it may be a problem that batch and farm effects will confound for some farms (where a farm has a single batch and therefore two random effect levels for a single observation).  We argue that slaughterhouse is the thing of interest here anyway and acknowledge the difficulty in separating farm and batch. This has been added to the Discussion

131 If you exclude the codes due to a significant random slaughterhouse effect you assume that farms go to at least 2 or more slaughterhouses in a balanced way. Did you check this assumption? Eg. if all bad farms only go to slaughterhouse A and all good farms only to slaughterhouse B, there will be almost no farm effect but a huge slaughterhouse effect… If both good and bad farms each go to slaughterhouse A and B there will be a huge farm effect and almost no slaughterhouse effect…

AU: Farms probably don’t send to slaughterhouses in any kind of balanced way, but we have no possible means to estimate this at the moment.  We hope to be able to look into this in the future, but for now we have to accept that we can’t differentiate low slaughterhouse sensitivity from a slaughterhouse where everyone sends the healthy animals, i.e. we assume that the distribution of true disease is random between slaughterhouses (which is likely nonsense due to spatial effects of disease prevalence).

We have added this aspect to the discussion

139 in my opinion corrections should always be done on the logit scale!

AU: We agree, but have added the reason why we include this information

139 Did you consider a model with slaughterhouse as fixed effect to asses significance of the slaughterhouse effect. You can still nest farm and batch within slaughterhouse as random component. Using such a model you can easily detect the biggest differences and conclude if correction is needed or not.

AU: Yes this would be a valid approach but it makes it harder (we would argue impossible) to compare farm and abattoir variation, which was the main point of the paper

140 and smaller significant differences means no systematic difference? You talk about corrections, how will you perform than in practice? Or do you assume that differences less than 2 (on logit scale) can be neglected?

 Please state your opinion in these differences

Are there inclusion criteria for batch or are all batches included?

AU: Large differences would need to be reduced through training of abattoir technicians.

Small differences might be corrected through the correction factors on the logit scale.

147-160: this information is already in M&M. please only refer to the tables

AU: We have rephrased to reduce the section to avoid repetition, but we do not agree that it is all Materials and Methods, while the specific codes excluded are a result of the exclusion process (described in materials and methods)..

166-176 idem

AU: As above

162-163: this is M&M move this to the appropriate part

AU: Done

177 I miss the number of farms, slaughterhouses and batches and the total number of animals included in this paper,

AU: The number of farms and slaughter houses are included in the text and the number of animals in Tables 1-4 

200 I really don’t like comparing variances of the random effect parts. But how did you test for the differences you state here? I am not yet convinced that 0.36 is bigger than 0.26 if you have a limited number of slaughterhouses, this will probably be borderline? Please also report test-statistics and p-values for stated differences.

AU: You are perfectly right that we cannot demonstrate that there is a significant difference between the random effects estimates, and it is tricky anyway because of the small number of slaughterhouses.  We have added information about how to interpret. No testing was done and the text has been rephrased

Fig 1 do S10-12 exclusively do sows? And the other only pigs? Very strange, this is uncommon in other countries.. Why random order of codes, this makes it difficult to find a specific code

AU: Yes, S10-S12 are exclusively sows. This has been clarified. The order of codes is not random. It is column-wise

217 Abattoir A71461? = Cx?

AU: This error has been corrected (all estimates are given) 

Fig 2 Why do you not integrate the calves and cows in one graph, this would be very interesting to see if differences are in the same line for calves and cows, they are slaughtered in the same slaughterhouses which make it more interesting to combine compared to Fig 1 where you did combine them although there were no slaughterhouses doing both types of animals. Same remark on ordering

AU: Because the same codes are not relevant for all cows and cows, and because they on top of each other so it would be difficult to demonstrate. They are actually over each other and ordered in the columns, so they can be compared

220 why do you expect an overall systematic difference for all codes the same, seems rather unlikely to me?

AU: If specific abattoirs had specific routines, this might result in systematic differences

254 how big is the group that exclusively send animals to only one slaughterhouse? Did you consider to exclude those farms in order to have a better estimation of the slaughterhouse-effect

AU: Most farms (overall: 82%) send to a single abattoir, so excluding those would change our population considerably and would thus not suit the purpose. We wanted this to include all Danish farms and throwing out most of them is a bad idea for this reason. An assumption of the approach is that there is no systematic difference in the true prevalence of disease at slaughterhouses (i.e. farms that send to >=2 abattoirs do so randomly, and farms that send to 1 abattoir choose this association randomly. Neither may be true as we discuss.

309-310 which is probably the case. Did you test how balanced division over slaughterhouses within the same farm was. If a large amount of farms go with a very small proportion of animals to one specific slaughterhouse this can be an indication;…

AU: We agree and have added additional discussion on this aspect

I miss the interpretation of farm-effect itself. If you have a very low slaughterhouse-effect it is a promising code in your opinion, but if also the farm-effect is very low, the added value for a WQ-index will be very limited in my opinion! 

AU No. These are observational data and balanced data is only something we could wish for. Therefore this discussion (which has been extended significantly)

Round 2

Reviewer 3 Report

nice impovement of the manuscript. Some of my concerns are not really solved, but it is probably not possible to do with the data you have available. But most of it is included in the discussion